# Biogasoline Obtained Using Catalytic Pyrolysis of *Desmodesmus* sp. Microalgae: Comparison between Dry Biomass and n-Hexane Extract

Noyala Fonseca [1], Roger Fréty [1,2] and Emerson Andrade Sales [1,3,*]

1   Laboratory of Bioenergy and Catalysis (LABEC), Polytechnic School, Federal University of Bahia—UFBA, Rua Aristides Novis, 2, 2nd Floor, Federação, Salvador CEP 40210910, Bahia, Brazil
2   Department of Physical Chemistry, Chemistry Institute, Federal University of Bahia, IQ/UFBA, Ondina University Campus, Salvador CEP 40170115, Bahia, Brazil
3   Industrial Engineering Post-Graduation Program (PEI), Polytechnic School, Federal University of Bahia—UFBA, Rua Aristides Novis, 2, 6th Floor, Federação, Salvador CEP 40210910, Bahia, Brazil
*   Correspondence: eas@ufba.br or andradesales.emerson@gmail.com; Tel.: +55-71-9-8887-8778

**Abstract:** The present work deals with the production of hydrocarbons in the C5–C12 range obtained from the fast micropyrolysis of a laboratory-grown *Desmodesmus* sp. microalgae. It compares the properties of this specific fraction of hydrocarbons using or not using transition alumina catalysts during pyrolysis in experiments with both pure dried microalgae and its n-hexane extract. The microalgae were characterised using thermogravimetry (TG) and CHN analysis; the n-hexane extract was analysed through Fourier transform infrared spectroscopy (FTIR). The pyrolysis experiments were performed in a multi-shot pyrolyser connected online with a gas chromatograph coupled to a mass spectrometer (GC/MS). The composition of the C5–C12 fraction was compared to that of an industrial pyrolysis gasoline. The results of pyrolysis at 600 °C show that the alumina catalyst increases the quantity of C5–C12 hydrocarbon families when compared to purely thermal pyrolysis, representing about 40% of all the dry microalgae pyrolysis products. In the case of n-hexane extract, the C5–C12 area fraction corresponds to 33.5% of the whole products' area when pyrolysis is conducted with an alumina catalyst. A detailed analysis shows that linear molecules, mainly unsaturated, are predominant in the products. Dry biomass formed more aromatic but less cyclic and alkylated molecules in relation to the n-hexane extract. Nitrogen products, essentially alkylated pyrroles, were produced in large quantities when dry biomass was used but were below the detection limit when pyrolysing the extracts. Thus, the extraction with hexane proved to be an effective way to remove nitrogen compounds, which are undesirable in fuels. The estimated low heating values of the present C5–C12 pyrolysis hydrocarbon fractions (between 43 and 44 MJ/kg) are quite comparable to the reported values for reformulated and conventional industrial gasolines (42 and 43 MJ/kg, respectively).

**Keywords:** *Desmodesmus* sp. microalgae; fast catalytic pyrolysis; n-hexane extract; biogasoline fraction

## 1. Introduction

Together with the finite amount of fossil fuels for the world population, their consumption in the foreseeable future is associated with environmental, economic and sustainability problems [1,2]. Raw biomass or extracted fractions of it seems to be an alternative, at least in part, for potential renewable feedstocks able to produce liquid biofuels, after some physical and chemical transformation [3]. Among non-edible raw biomass materials requiring limited land use, microalgae are increasingly studied as they are efficient in the synthesis of renewable fuel precursors such as polysaccharides and oily products. Polysaccharides after fermentation can produce alcohols as additives or substitutes for gasoline, whereas oily products can be used as a feed for hydrocarbon-based liquid fuels, after deoxygenation

and cracking. Microalgae transformation is therefore an impressive and challenging line of research implying progress in strain selection, production technology and harvesting and post-harvesting, as well as in microalgae transformation processes to increase renewable feed sources' potential and gain independence from fossil ones. Furthermore, the continuous evolution of the road and aviation energy sector, leading to frequent updates in fuel specifications, is a strong incentive for new studies in the field of renewable liquid fuels. In the specific case of gasoline, both hydrocarbons of fossil origin and hydrocarbons from renewable feeds must follow safety requirements associated with quality, an important factor in the production, control and handling, as well as motor lifetime [2,4].

Parallel to fuel performance requirements, the environmental projections are also very stringent. Thus, many works have focused on the production of biogasoline, biokerosene and other biofuels, minimizing the environmental impacts associated with land use, extractive processes, other chemical processes and final fuel use, proposing cleaner technologies and catalysts capable of producing fractions as close as possible to fossil fuels [5].

Microalgae are considered an alternative to lignocellulosic biomass, capable of being a source of bio-oils and biofuels. Biogasoline can be produced by mixing microalgae biofuels with conventional petroleum products to provide the necessary specification properties in the form of "drop in" fuels [6,7] However, the presence of proteins in microalgae, a source of nitrogen products during the conversion of biomass, can be considered a problem, as nitrogen compounds are potential poisons for catalysts used in refining complex hydrocarbon mixtures and are a source of NOx emission, a strong air pollutant, when nitrogenated molecules are burned in engines [8].

Some ways to limit the nitrogen content in potential energy feedstocks produced by microalgae are (i) preliminary thermal treatment at temperatures close to 200 °C [9] and use of the resulting solid residue as feedstock; and (ii) selective chemical extraction of the lipids and the use of this fraction as feed for liquid fuels. The extraction of microalgae oily molecules is effective in the simplification of the matrix and reduces the content of nitrogen compounds [10–14] although also reduces the overall yield of fuel.

Microalgae strains are important in wastewater treatment [15–17], and among them, *Desmodesmus* sp. has emerged as the most promising strain, removing almost all nitrogen and phosphate from effluents. Its biomass presents a calorific value similar to terrestrial plants and also has potential for use as a biolubricant. Therefore, *Desmodesmus* sp. shows promise for wastewater treatment, energy and industrial applications. According to [18], if such an application progresses, more *Desmodesmus* microalgae will be available as biomass feed for energy purposes.

Pyrolysis studies on *Desmodesmus* sp. microalgae grown at laboratory scale have been performed by [19,20] in absence of any catalyst and by [21] on the bio-oil resulting from hydrothermal treatment. In the first case, depending on the cultivation conditions, it was found that pyrolysis temperatures between 600 and 700 °C were necessary to obtain a satisfactory yield of hydrocarbons, whereas in the second case, it was observed that the bio-oil from microalgae in the presence of an HZSM5 catalyst produced, at a pyrolysis temperature of 600 °C, a hydrocarbon yield between that of the cellulose and hexatriacontane feeds.

The present study used both dried *Desmodesmus* sp. microalgae and its n-hexane extract, focusing on the production of biogasoline, i.e., the formation of a C5–C12 hydrocarbon fraction. The catalytic pyrolysis was performed in the presence of $\gamma$-Al$_2$O$_3$ as a reference catalyst, used in the past by different groups to efficiently deoxygenate oily molecules from other sources through catalytic pyrolysis [14,22–25]. In the present work, extraction with hexane proved to be an effective way of removing undesirable nitrogen compounds in fuels.

## 2. Results and Discussion

### 2.1. Thermogravimetry

Figure 1 shows the results of the thermogravimetric analysis of *Desmodesmus* sp. microalgae from room temperature to 650 °C, under pure nitrogen flow: % of mass loss—TG (red curve); differential thermal gravimetry—DTG (blue curve) and differential thermal analysis—DTA (black curve). The thermogravimetric analysis of *Desmodesmus* sp. microalgae under synthetic air flow was performed by our group and presented previously [14]. Three main mass losses I, II and III were observed in both cases, although the temperature limits were slightly different. Table 1 summarizes the mass losses obtained, in percentages, for the three main events, under both synthetic air and $N_2$ flows, and the ash or ash + char quantity. The results of the elementary analysis are also given in the right part of Table 1. The values obtained for the C and N contents are in satisfactory agreement with the literature data [14,26].

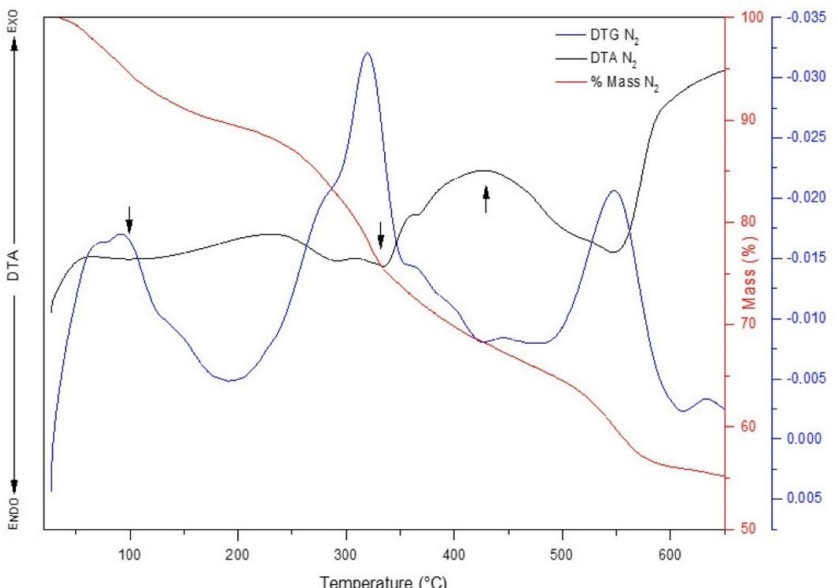

**Figure 1.** Thermogravimetric analysis of *Desmodesmus* sp. microalgae biomass under nitrogen gas flow (TG: red curve, DTG: blue curve, DTA: black curve).

**Table 1.** Percentage of mass loss for events (I) (water), (II + III), volatiles under $N_2$ flow and ash + char, C, H and N contents of *Desmodesmus* sp. biomass.

| Atmosphere | Event I up to 200 °C (%) | Event II 200–450 °C (%) | Event III above 450 °C (%) | Ash + Char (%) | % C | % H | % N | % Others |
|---|---|---|---|---|---|---|---|---|
| Synthetic air | 8 | 48 | 33 | 11 | 43.0 | 13.8 | 6.4 | 36.8 |
| Nitrogen | 10 | 22 | 17 | 55 | | | | |

Event I, ended at about 200 °C, presented one mass loss, associated with a small endothermic phenomenon attributed to dewatering. Event II (between 200 and 450 °C) showed an important mass loss, with a maximum rate loss at 330 °C. This event is complex, as the corresponding DTA trace starts with an endothermic phenomenon followed by a complex exothermic one ending at 420 °C. These observations suggest that together with the progress of endothermic devolatilization (proteins and light fatty molecules), exothermic phenomena appear, associated with the beginning of molecule oxidation. As the gas atmosphere is very poor in oxygen, and as the DTG and DTA are not fully synchronized (the extremum of the DTA is slowed compared to the extremum of the DTG), it is thought that the exotherms are essentially due to auto-oxidation, i.e., the oxygen contained in some molecules is reacting with neighbour H and C atoms to form $H_2O$ and $CO_2$. However, as

the content of structural oxygen is limited, together with the transfer of cracked molecules in the gas phase, some carbon-rich residues may also form. In fact, as shown in Table 1, the mass loss during Event II (22%) is much lower compared with the mass loss observed under a synthetic air atmosphere (48%) [14].

Event III (above 450 °C) is linked to the decomposition of fatty molecules with high molecular weights, together with further auto-oxidation of some heavy molecules and of their carbon residues. In that case, the extrema of the mass loss rate and of the DTA exothermic signal are practically similar. The temperature of the maximum heat release, 560 °C, is close to the temperature at which the carbon deposited on the catalysts after hydrocarbon cracking is burnt under the oxygen medium [27,28]. The final mass of the *Desmodesmus* sp. after TG under $N_2$ flow is 55% of the initial mass, much higher than the final mass obtained under air flow (11%); this difference is attributed to ash enriched with carbon deposits when TG is performed under a $N_2$ atmosphere. Although in both TG situations the exact chemical state of ash is not known, it appears that in conditions of slow pyrolysis or TG under nitrogen flow, the residual microalgae mass is composed of minerals salt precursors of ash and of char, i.e., a solid phase rich in minerals and carbon.

### 2.2. Fourier Transform Infrared Spectroscopy (FTIR)

Figure 2 presents the FTIR spectrum of the n-hexane extract of *Desmodesmus* sp. This extract represents only ca. 2.5 wt.% of dried and ash-free microalgae. A large band at 3450–3500 cm$^{-1}$ is due to OH groups (fatty acids and residual water). The bands at 2928 and 2853 cm$^{-1}$ are associated with the stretching modes of C-H existing principally in the carbon chain of fatty components; the bands close to 1746 and 1160 cm$^{-1}$ are attributed to the C=O and C-O-C stretching vibrations of fatty acids and esters, respectively. The bands at around 1381 and 1457 cm$^{-1}$ are associated with rotation vibrations of $CH_2$ and $CH_3$ functional groups, whereas the poorly resolved band close to 1647 cm$^{-1}$ is tentatively attributed to C-O stretching vibration in amide. To summarize, the FTIR spectrum of the n-hexane extract of *Desmodesmus* sp. suggests that this extract is mainly composed of triacyl glycerides and derived molecules, in agreement with previous results [29,30].

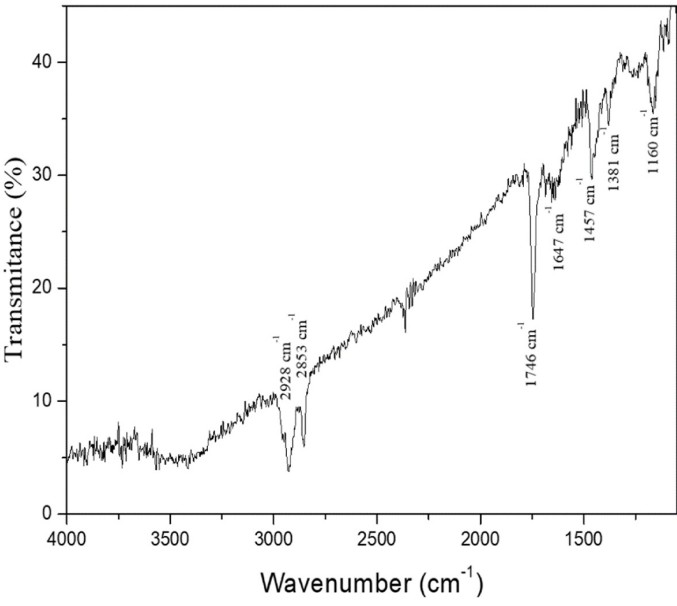

**Figure 2.** Fourier transform infrared spectrum (FTIR) of the n-hexane extract of *Desmodesmus* sp. microalgae.

### 2.3. Pyrolysis of the Microalgae Biomass and n-Hexane Extract

Figure 3 presents the pyrograms obtained in the presence of alumina catalyst for the dry *Desmodesmus* sp. and *Desmodesmus* sp. n-hexane extract, as well as the pyrogram fractions between retention times 2.0 and 9.5 min after pyrolysis at 600 °C. The pyrograms

obtained in the absence of alumina showing similar types of profiles are not displayed, although their profile between the C5–C12 carbon number molecules was analysed.

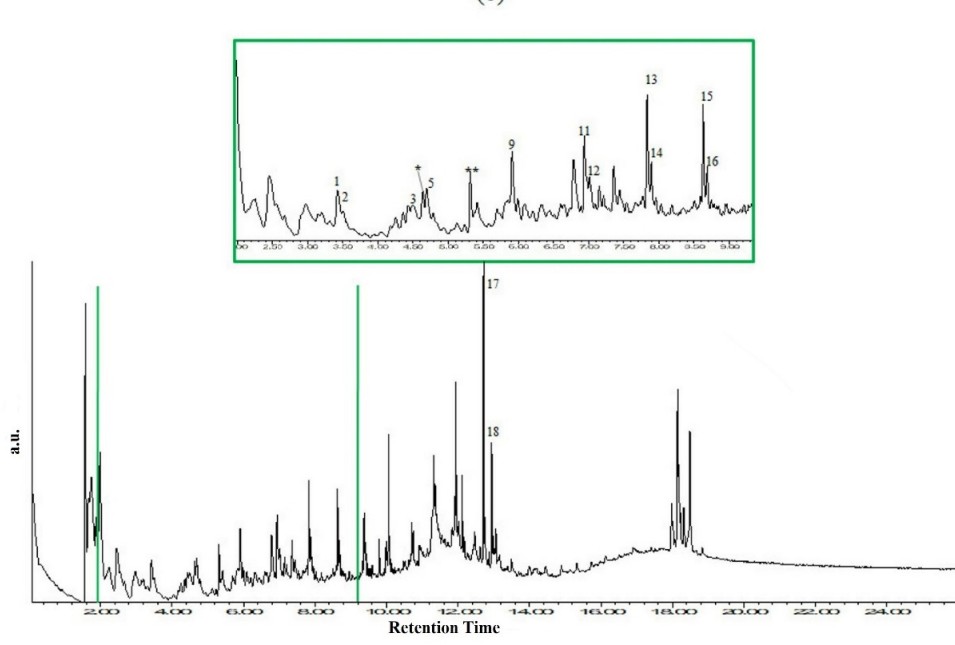

**Figure 3.** Complete pyrogram and pyrogram fragment between retention times 2.0 and 9.5 min (defined with two vertical green lines in the complete pyrogram) after pyrolysis at 600 °C. (**a**) Dry *Desmodesmus* sp. with alumina catalyst; (**b**) n-hexane extract of *Desmodesmus* sp. with alumina catalyst. Identification of some products in the complete and limited fragment pyrograms: (1) 1-Heptene; (2) n-Heptane; (3) Toluene; (4) 1-Octene; (5) n-Octane; (6) 1H-Pyrrole, 3-methyl-; (7) 1H-Pyrrole, 2-methyl-; (8) Ethylbenzene; (9) 1-Nonene; (10) Styrene; (11) 1-Decene; (12) n-Decane; (13) 1-Undecene; (14) n-Undecane; (15) 1-Dodecene; (16) n-Dodecane; (17) 3-Octadecyne; (18) Hexadecanal; (19) Pentadecanenitrile; (*) 1-Heptene, 2-methyl-; (**) 2,4-Dimethyl-1-hexene.

The main products obtained in all cases are alkenes and alkanes. 1-alkenes are predominant when pyrolysis is conducted in the absence of a catalyst as was reported previously during a pyrolysis study of pure saturated and unsaturated fatty acids [31,32], but iso-alkenes were also observed in the presence of the alumina catalyst, resulting in agreement with the studies of [24,33,34]. Such results confirm the presence of fatty compounds in the *Desmodesmus* sp. biomass and biomass extract. However, considering the whole pyrograms, it is also apparent that, together with simple hydrocarbon-based molecules, alkynes and oxygenated and nitrogenated products are obtained during the pyrolysis. Linear Octadecyne, Hexadecanal and Pentadecanenitrile are the products of higher content when pyrolysing dry microalgae, but Hexadecanal is also present in a small amount in the case of pyrolysis of the *Desmodesmus* sp. n-hexane extract.

Table 2 presents the percentage of hydrocarbons and nitrogenated and oxygenated compounds in the C5–C12 range obtained after pyrolysis, presented in a pseudoquantitative way, i.e., using the % area of the sum of the product family over the total area of the pyrogram.

**Table 2.** Pyrolysis at 600 °C of *Desmodesmus* sp. microalgae and of its n-hexane extract in presence or absence of alumina catalyst: semiquantitative estimation of the C5–C12 fraction in the whole pyrogram, of the hydrocarbon fraction and of the oxygenated and nitrogenated products, and the sum of oxygenated, nitrogenated and unidentified groups in the complete pyrogram.

| | Alumina | % C5–C12 Products in the Whole Pyrogram | % C5–C12 Hydrocarbons in the Whole Pyrogram | % Nitrogenated Compounds in the C5–C12 Fraction | % Oxygenated Compounds in the C5–C12 Fraction | % Other C5–C12 Products in the Whole Pyrogram |
|---|---|---|---|---|---|---|
| Dry biomass | no | 18.5 | 13.7 | 0.5 | 0.7 | 4.8 |
| Dry biomass | yes | 48.7 | 40.4 | 6.4 | 0.2 | 8.3 |
| Extract | no | 24.9 | 24.4 | b.d.l. * | b.d.l. * | 0.5 |
| Extract | yes | 44.6 | 33.5 | b.d.l. * | 2.6 | 11.1 |

\* b.d.l. = below detection limit.

From Table 2, some comments can be made: (i) the C5–C12 product fraction is always higher in the presence of alumina catalysts than in their absence for both biomass and its n-hexane extract, confirming that the alumina catalysts have induced a rather important cracking activity in both cases; (ii) the C5–C12 hydrocarbon amount is always more important in the presence of alumina catalyst than in its absence, confirming that alumina also exerts clear deoxygenation activity on both biomass and its extract, as observed previously [24]; (iii) the role of the alumina catalyst is more visible with pure *Desmodesmus* sp. biomass than with its n-hexane extract, and such a difference is tentatively associated with the fact that in the case of dry biomass pyrolysis, the catalyst bed is located above the biomass and is susceptible to help transform all the gaseous products of thermal pyrolysis of the *Desmodesmus* sp. sample, and such bed geometry was not obeyed in the case of n-hexane extracts simply dropped either in the reacting cup or in the reacting cup containing a defined quantity of alumina; (iv) the quantity of other C5–C12 products, generally associated with oxygenated and nitrogenated products, is always higher in presence of alumina; then, the alumina catalyst is helping the production of the desired pure hydrocarbons but also favours the formation of more complex molecules, in particular nitrogenated molecules and to a lesser extent some oxygenated products; (v) the nitrogenated products, essentially alkylated pyrroles, were produced to a rather large extent with the dry biomass but were under the detection limit when pyrolysing the extracts. This is an important result, as the reduction of the content of nitrogen compounds is a goal for increasing the quality of microalgae pyrolysis products [35].

The quality of the biogasoline produced was estimated through the separation of the hydrocarbon fraction into linear, cyclic + alkylated molecules, independently of their saturated or unsaturated form (simple hydrogenation being sufficient to saturate alkenes,

cycloalkenes and polyunsaturated compounds) and aromatics. The results are presented in Table 3 as area % in reference to the total area of the pyrograms.

**Table 3.** Semiquantitative estimation of the type of molecules (linear, cyclic + alkylated, and aromatics) of the C5–C12 fraction after pyrolysis at 600 °C of *Desmodesmus* sp. microalgae and of its n-hexane extract, in presence or absence of alumina catalyst.

| Sample (*Desmodesmus* sp.) | Alumina Catalyst | Type of Molecules (% Area) | | |
|---|---|---|---|---|
| | | Linear | Cyclic and Alkylated | Aromatics |
| Dry biomass | no | 9.4 | 1.3 | 3.1 |
| Dry biomass | yes | 25.3 | 2.4 | 14.6 |
| Extract | no | 15.5 | 8.5 | 1.3 |
| Extract | yes | 20.7 | 4.9 | 2.8 |

Table 3 shows that the hydrocarbons in their linear form always present a higher quantity than the other molecule types, cyclic + alkylated and aromatics, suggesting that consecutive transformations of cracked molecules or radicals are not playing a definite role in the present experimental conditions. The cyclic and alkylated molecules appear in a larger amount when pyrolysing *Desmodesmus* sp. extract compared to the pyrolysis of dry microalgae, whereas the formation of aromatics followed an opposite trend. Such a situation is beneficial to the production of a gasoline fraction with better combustion properties. The higher production of aromatics using dry biomass may have two distinct origins: on one hand, proteins' presence in the dry biomass can be considered a potential precursor molecule for aromatics, essentially in the presence of acid catalyst [26,27,36]; on the other hand, the fact that the alumina catalyst layer is above the layer of the microalgae biomass may help consecutive transformations, allowing alkylated and cyclic molecules to be further transformed into more dehydrogenated molecules as intermediates to aromatic formation.

Finally, the distribution of the C5-C12 fraction as a function of carbon chain length is presented in Figure 4.

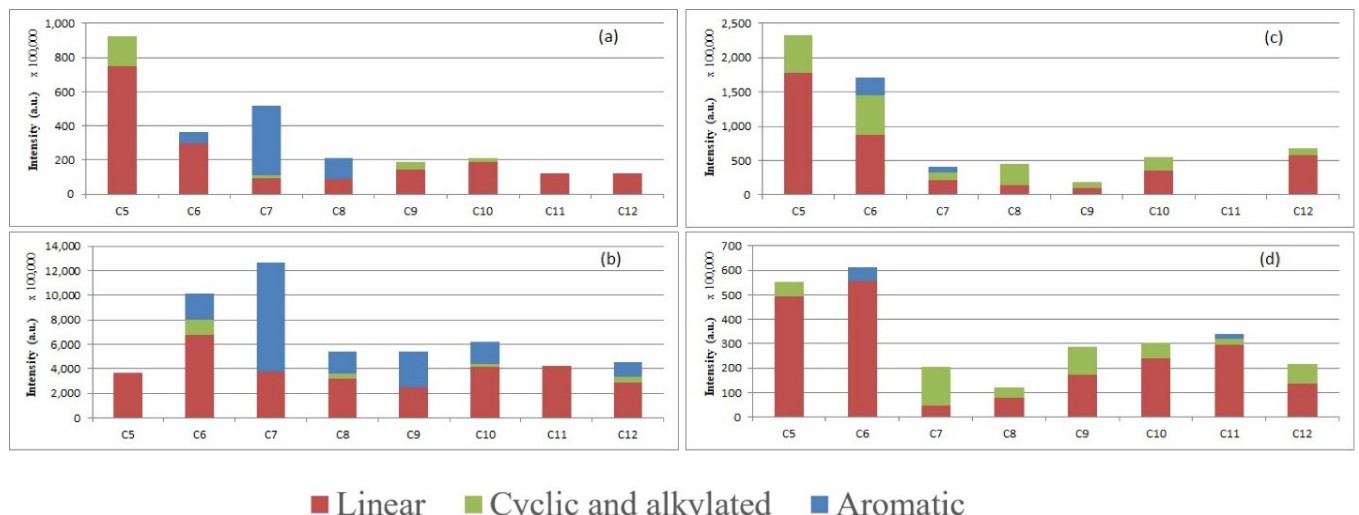

**Figure 4.** Chemical distribution of the main hydrocarbon families between C5–C12 after pyrolysis at 600 °C: (**a**) dry biomass without catalyst, (**b**) dry biomass with alumina catalyst, (**c**) n-hexane extract without catalyst and (**d**) n-hexane extract with alumina catalyst.

Light hydrocarbons (C5–C7) are always in a larger proportion than the heavier ones. The pyrolysis in the presence of alumina enabled obtaining a higher proportion of C8–C11 hydrocarbons, which may be interesting to decrease the volatility of the present pyrolysis

fraction. The low heating values of the present C5–C12 pyrolysis hydrocarbon fractions were estimated from their composition, based on reported data [37–39], and are shown in Table 4.

**Table 4.** Estimated low heating values (MJ/kg) of the C5–C12 fraction after pyrolysis at 600 °C of *Desmodesmus* sp. microalgae and of its n-hexane extract, in presence or absence of alumina catalyst.

| Sample (*Desmodesmus* sp.) | Alumina Catalyst | Mean Low Heating Value (MJ/kg) |
|---|---|---|
| Dry biomass | no | 44 |
| Dry biomass | yes | 43 |
| Extract | no | 44 |
| Extract | yes | 43 |

The values presented in Table 4 (between 43 and 44 MJ/kg) are quite comparable to the reported values for reformulated and conventional industrial gasolines (42 and 43 MJ/kg, respectively) [37–39].

Figure 5 shows the repartition of hydrocarbon compound families in an industrial pyrolysis gasoline sample analysed with the same equipment for comparison with the actual pyrolysis results, using both dried *Desmodesmus* sp. microalgae and its n-hexane extract (Figure 4). The C5–C12 fraction, whatever the starting material (dry microalgae or n-hexane extract), is on one hand too poor in C8–C9 product content compared to the industrial sample, but on the other hand it is rich in linear unsaturated products. Therefore, before using such fractions as "drop in fuel", a less extensive cracking and partial hydrogenation will be necessary. Modifications to the pyrolysis conditions and to the type of catalyst are under progress. Other topics that could be the subject of future studies include extraction using green solvents, preliminary thermal treatments such as torrefaction to limit the amount of nitrogenated compounds in dry biomass, and the use of hydrogen-assisted pyrolysis. Regarding the reaction itself, metal catalysts supported on alumina and the possibility of a composite or multilayer catalyst bed could be explored in the future.

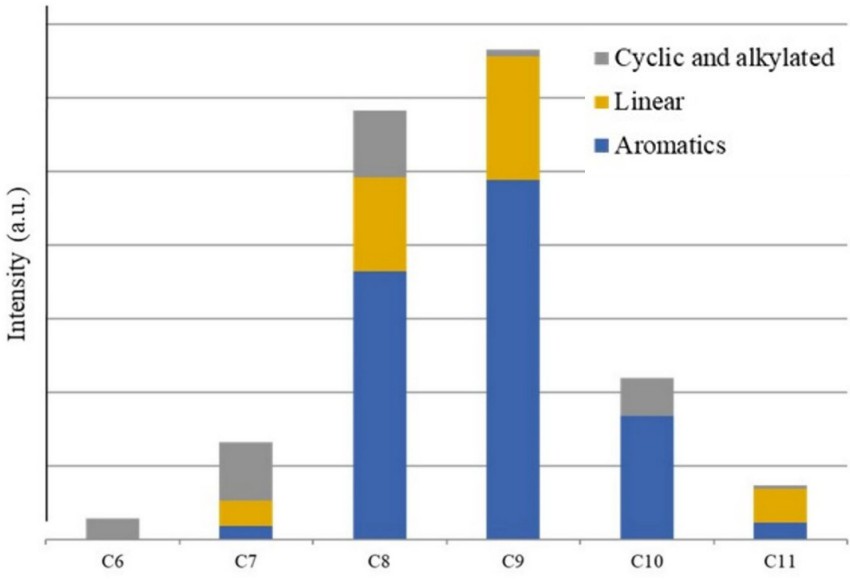

**Figure 5.** Distribution of hydrocarbon families in "industrial pyrolysis gasoline".

## 3. Materials and Methods

### 3.1. Reagents

Chemical extraction was performed with n-hexane PA (Sigma Aldrich, St. Louis, MO, United States), following the methodology described previously [14].

The catalyst used in the present work was a γ-alumina (Alcoa Alumínio S.A., Poços de Caldas, MG, Brazil), Code: A1. It contains 0.013% silica and 0.48% sodium and presents a BET surface area of 71 m$^2$/g. It also showed acidity estimated through ammonia thermo-desorption.

A commercial pyrolysis gasoline A type, obtained from "Distribuidora de combustíveis BR S/A", Salvador, Bahia, Brazil, was used as a reference product and did not contain any ethanol, which is always added in various amounts to common gasoline in Brazil.

### 3.2. Cultivation of the Microalgae

Microalgae cultivation was performed in 1 L conical flasks in a phototrophic batch, with cultivation media that tried to use the main nutrients needed for the species to develop under conditions comparable to those found in the nature. The *Desmodesmus* sp. species, being natural from freshwater, were cultivated using Bold's Basal Medium (BBM) in distilled water. The general conditions of the cultivation were described previously [14]. The harvesting occurred after a 40-day culture; the culture medium was centrifuged (4200 rpm for 10 min) for the collection of the cells and the supernatant solution was discarded. The resulting wet biomass was dispersed in Petri plates and dried in an aerated oven at 60 °C for 24 h, before being stored at −10 °C. The dried microalgae were used to obtain oily compound extracts, using n-hexane as a solvent.

### 3.3. Thermogravimetric Analyses of Biomass

Thermogravimetric analyses were conducted to determine the amounts of highly volatile material (including moisture), mildly volatile material (volatiles) and ash content. These analyses were carried out on a DTG-60/60H thermo-balance (Shimadzu, Kyoto, Japan), using a dried microalgae sample mass of approximately 2 mg, a platinum sample holder and a heating rate of 20 °C/min. Thermograms were obtained either under synthetic air or pure nitrogen flows (10 mL/min, between 20 °C and 650 °C).

### 3.4. Elementary Characterisation (CHN)

The biomass of the dry microalgae was characterised in terms of carbon, hydrogen and oxygen content following the ASTM D 529111 method. The setup was the Perkin Elmer 2400 series (Waltham, MA, United States), based on the Pregl-Dumas method, in which 1.0 mg of dried sample was submitted to combustion in an atmosphere of pure oxygen. The resulting gases of this combustion were quantified using a thermal conductivity detector (TCD).

### 3.5. Fourier Transform Infrared (FTIR) Spectroscopy of n-Hexane Extract

The dry n-hexane extract, after mixing with KBr, was pelletised and analysed with infrared spectroscopy using a Bomen spectrophotometer model MB 102 (New South Wales, Australia), at room temperature (23 °C) under an ambient atmosphere.

### 3.6. Pyrolysis and Product Analysis

Pyrolysis reactions were performed under a helium atmosphere using a Frontier Laboratories LTD Multi-Shot Pyrolyzer Model EGA/PY-3030D (Fukushima, Japan) connected online with a gas chromatograph coupled to a mass spectrometer (Agilent GC-MS 5799A, Santa Clara, CA, USA)). The main products were separated with a Frontier Laboratories UA5-30M-0.25F GC column (30 m length, 0.25 mm diameter, 0.25 μm film thickness, 5% diphenyl and 95% dimethyl polysiloxane stationary phase), subjected to an initial temperature of 40 °C for 2 min, followed by a heating ramp at a rate of 20 °C/min up to 320 °C, with the temperature maintained for 10 more min. The MS ion source and the interface pyrolyzer/injector temperatures were both fixed at 320 °C. The values of *m/z* were set in the range of 40 to 400, in scan mode. The helium flow in the column was 1 mL/min, and the split ratio was 1/50.

For the analysis of the products, the chromatographic peaks of the pyrograms with an area lower than 0.1% of the total area were neglected. The peaks were identified using the NIST Database, and only those products with an identification probability greater than 70% were considered. Products presenting an incoherent identification in terms of logic in the appearance sequence and those with a probability of identification below 70% were grouped as unidentified—NI. The estimated compositions after pyrolysis were calculated by dividing the peak areas of the specific products by the total area of the non-neglected peaks, including the unidentified products.

The pyrolysis method consisted of dropping under the helium flowing atmosphere the sample holder (Eco-cup LF PY1-EC80F, Frontier LAB) in the pyrolysis area preheated to 600 °C and waiting for 15 s before a gas aliquot was injected in the online analytical system. Samples to be pyrolysed were inserted in the pyrolysis sample holder in different ways. In the case of thermal pyrolysis, the dry extract obtained from 200 mg of dry biomass, according to the procedure described in [14,22], was dissolved in 200 µL of n-hexane, and a quantity corresponding to 0.02 mg of extract was deposited into the sample holder using a microsyringe; the pyrolysis cup was inserted into the setup, purged with helium for the time necessary to completely vaporise the solvent and then dropped in the preheated reacting area. For catalytic pyrolysis, 0.2 mg of alumina was first deposited into the cup; then n-hexane containing a known amount of extract was poured on the alumina layer and the cup was inserted in the setup and purged with helium; the time necessary to vaporize the n-hexane solvent was obeyed, and then the sample cup was dropped in the preheated reacting area.

For pyrolysis without a catalyst, in the case of the dry microalgae, a 2 mg sample was simply poured into the cup before proceeding as before. For catalytic pyrolysis, a layer of alumina was added on top of the microalgae before inserting the cup and its samples into the equipment.

In all cases, a tiny layer of quartz wool was deposited in the cup, on top of the samples.

The industrial pyrolysis gasoline analysis was performed in the pyrolysis setup described previously; 5 µL of gasoline was poured into the sample pan and was dropped in the reacting area preheated to 350 °C. All other parameters were maintained as before.

The uncertainty of the measurements of the pyrolysis experiments as a whole, including from sample preparation to chromatographic analysis, was evaluated by running duplicates performed by the same operator, and the average percent standard deviation for some classes of compounds was calculated, as shown in Table 5 below:

**Table 5.** Standard deviation calculated for some types of hydrocarbons detected in the pyrolysis experiments.

| Type of Hydrocarbon | Standard Deviation (%) |
|---|---|
| Alkane | 7.3 |
| Alkene | 2.7 |
| Aromatic | 2.1 |
| General average | 5.7 |

The results of the standard deviations obtained for these specific types of compounds demonstrate a high reliability in the results presented, especially considering that the work involves a biomass.

## 4. Conclusions

The present study shows that the fast pyrolysis at 600 °C of *Desmodesmus* sp. microalgae or/and its n-hexane extract, in the absence or presence of a transition alumina catalyst, can be considered a way to produce a precursor of "drop in" gasoline. The alumina catalyst increased the quantity of C5–C12 hydrocarbon compounds when compared to purely thermal pyrolysis, representing about 40% of all the dry microalgae pyrolysis products and 33.5% of the products when using the n-hexane extract. A detailed analysis shows

that linear molecules, mainly unsaturated, are predominant in both cases. Dry biomass formed more aromatics but less cyclic and alkylated molecules in relation to the n-hexane extract. Extraction with hexane proved to be an effective way to remove undesirable nitrogen compounds in fuels, as nitrogenated products, essentially alkylated pyrroles, were produced to a rather large extent with dry biomass but were under the detection limit when pyrolysing the extracts. Compared to industrial pyrolysis gasoline, the C5–C12 fraction that was obtained always contained a quite limited quantity of C8–C9 molecules. This "drop in" gasoline fraction is not fully comparable to industrial pyrolysis gasoline, and its composition could probably be optimised using other types of catalysts to form more isomerised and hydrogenated products at the end of pyrolysis without increasing too much the formation of aromatics. Despite this fact, the estimated low heating values of the present C5–C12 pyrolysis hydrocarbon fractions (between 43 and 44 MJ/kg) are quite comparable to the reported values for reformulated and conventional industrial gasolines (42 and 43 MJ/kg, respectively). The selection of microalgae containing a higher content of oily components, with less polyunsaturated compounds, is probably another way to further optimize the quality of biogasoline issued from the fast pyrolysis of microalgae. Strategies to reduce the content of nitrogen compounds, such as the one presented here, have another objective to increase the quality of microalgae pyrolysis products.

**Author Contributions:** Conceptualization, N.F.; methodology, N.F., R.F. and E.A.S.; validation, N.F., R.F. and E.A.S.; formal analysis N.F., R.F. and E.A.S.; investigation, N.F., R.F. and E.A.S.; resources, R.F. and E.A.S.; data curation, N.F., R.F. and E.A.S.; writing—original draft preparation, N.F. and R.F.; writing—review and editing, N.F., R.F. and E.A.S.; supervision, R.F. and E.A.S.; project administration, E.A.S.; funding acquisition, E.A.S. All authors have read and agreed to the published version of the manuscript.

**Funding:** This research was funded by the Bahia State Research Support Foundation (FAPESB), Higher Education Personnel Improvement Coordination (CAPES) and the National Council for Scientific and Technological Development (CNPq)—Brazil, grant number 407880/2017-8.

**Acknowledgments:** The authors are grateful to the Bahia State Research Support Foundation (FAPESB), Higher Education Personnel Improvement Coordination (CAPES) and the National Council for Scientific and Technological Development (CNPq)—Brazil. Furthermore, the authors are grateful to the following laboratories of the Federal University of Bahia (UFBA)—Brazil: Coordination Chemistry Research Group (GPQC) for the assistance in the FTIR analysis and the Catalysis and Environment Laboratory (CATAM) for the thermogravimetric analyses.

**Conflicts of Interest:** The authors declare no conflict of interest.

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
