# Peer review of "Biogasoline Obtained Using Catalytic Pyrolysis of Desmodesmus sp. Microalgae: Comparison between Dry Biomass and n-Hexane Extract"

_catalysts, doi:10.3390/catal12121517_

Round 1

Reviewer 1 Report

The manuscript by Fonseca et al. described catalytic pyrolysis of Desmodesmus sp. microalgae for biogasoline production. They investigated the effect of n-hexane extraction of microalgae and alumina catalyst on product yields and distributions. Although some data are interesting, the manuscript requires more data for complete comparison and in-depth discussion. The followings are some comments to improve the manuscript.

(1)   Figure 1 shows only TG, DTG and DTA data under nitrogen gas flow. It is required to show the data under air flow, as compared in Table 1.

(2)   Figure 2 shows only the FTIR spectrum of n-hexane extract of the sample. For comparison, it is required to show the FTIR data of the microalgae.

(3)   Figure 3 shows the pyrograms after pyrolysis of two sample. It is required to show the Product distributions with yields and selectivities in Table for clear comparison.

(4)   Higher heating values of the products should be shown also, as it is targeted for biogasoline production.

(5)   English should be checked more carefully.

Author Response

Reviewer 1

The authors thank Reviewer 1 for his comments and suggestions. The five points will be answered thereafter, and eventual modifications added in the new submission.

(1)   Figure 1 shows only TG, DTG and DTA data under nitrogen gas flow. It is required to show the data under air flow, as compared in Table 1.

Experiments of TG, DTG and DTA under air flow of Desmodesmus sp. microalgae were realized by our research group and the data were shown previously in ref [14] and the authors thought unnecessary to present two times the same data in different publications.

(2)   Figure 2 shows only the FTIR spectrum of n-hexane extract of the sample. For comparison, it is required to show the FTIR data of the microalgae.

Point 2: The obtention of FTIR spectra of pure Desmodesmus sp. microalgae biomass was performed only one time. But the obtained spectrum with the available equipment was very complex, with low resolution, and it was decided to cancel such measurements due to our limited experimental conditions. However, in a recent publication (https://doi.org/10.1016/j.biortech.2022.127445), FTIR spectra of Nordic Desmodesmus microalgal strain were presented in the supplementary data, between 800 and 1900 cm-1. Intense bands were observed at around 1550 and 1650 cm-1, much more intense than that attributed to ν(C=O) stretching (~1750 cm-1) due to esters and fatty acids. Then, the spectrum of Desmodesmus extract presented in Figure 2 can be considered as “amide free” and confirmed that the present n-hexane extraction was practically 100% selective for fatty acids and esters.

(3)   Figure 3 shows the pyrograms after pyrolysis of two sample. It is required to show the Product distributions with yields and selectivities in Table for clear comparison.

The data presented in Tables 2 and 3 were not modified: fast micro pyrolysis results cannot be treated as results from a true catalytic reactor: two different product families are formed i.e., those produced inside the solid bed (primary products) and those formed in the gaseous phase around the catalytic bed and during the transport to the chromatograph, in a space where the temperature was not quite homogeneous. Therefore, yields and selectivity are too dependent on equipment geometry and in the present conditions, only trends can be proposed. We present below the absolute and relative peak intensities for the compounds cited in Fig.3, for quantitative comparison between them.

Compunds cited in Fig. 3

(a) dry Desmodesmus sp., with alumina catalyst

(b) n-hexane extract of Desmodesmus sp., with alumina catalyst

absolut area

relative area

absolut area

relative area

 (1) 1-Heptene

206912348

1

5323384

1,0

(2) n-Heptane

171204523

0,8

2507789

0,5

(3) Toluene

886663313

4,3

2705539

0,5

(4) 1-Octene

175578957

0,8

-

-

 (5) n-Octane

114130141

0,6

6578495

1,2

(6) 1H-Pyrrole, 3-methyl-

60181225

0,3

-

-

(7) 1H-Pyrrole, 2-methyl-

91487137

0,4

-

-

(8) Ethylbenzene

85137140

0,4

-

-

(9) 1-Nonene

161597779

0,8

2366947

0,4

 (10) Styrene

26240212

0,1

-

-

(11) 1-Decene

228604276

1,1

1030795

0,2

(12) n-Decane

104702129

0,5

12670736

2,4

(13) 1-Undecene

198522643

1,0

1630484

0,3

(14) n-Undecane

107263100

0,5

14966663

2,8

 (15) 1-Dodecene

167463698

0,8

598578

0,1

(16) n-Dodecane

95246442

0,5

8483743

1,6

(17) 3-Octadecyne

570897090

2,8

34363942

6,5

 (18) Hexadecanal

266106308

1,3

16564949

3,1

(19) Pentadecanenitrile

388978718

1,9

-

-

 (*) 1-Heptene, 2-methyl-

-

-

4174184

0,8

(**) 2,4-Dimethyl-1-hexene

-

-

1976004

0,4

(4)   Higher heating values of the products should be shown also, as it is targeted for biogasoline production.

We agreed with Referee 1, and we calculated the heating values of some specific molecules of the C5-C12 fraction, based on the literature data. These values were compared to heating values of some gasolines: the results were summarized in Table 4, and suggest that the C5-C12 fractions obtained in the present work can be considered as excellent “drop in” mixtures. We added now in the text:

. The low heating values of the present C5-C12 pyrolysis hydrocarbon fractions were estimated from their composition, based on reported data [37-39] and are shown in Table 4.

Table 4. Estimated low heating values (MJ/kg) of the C5-C12 fraction after pyrolysis at 600°C of Desmodesmus sp. microalgae and of its n-hexane extract, in presence or absence of alumina catalyst.

Sample (Desmodesmus sp.)

Alumina catalyst

Mean low heating value (MJ/kg)

Dry biomass

no

44

Dry biomass

yes

43

Extract

no

44

Extract

yes

43

The values presented in Table 4 (between 43 and 44 MJ/kg) are quite comparable to the reported values for reformulated and conventional industrial gasolines (42 and 43 MJ/kg, respectively) [37-39].

(5)   English should be checked more carefully.

The authors made a fine reading of their new submitted text and made few modifications, as shown in the text, including in the References Sectioon._____

Reviewer 2 Report

The Manuscript ID: catalysts-2010467 Biogasoline obtained by catalytic pyrolysis of Desmodesmus sp. microalgae: comparison between dry biomass and n-hexane extract.” requires revision before accepted for publication. The specific comments are given below.

1.     Abstracts needs to have more precision as in the current form it appears. In the abstract, please add an indication of the achievements from your study that is relevant to the journal scope. Please also add a research background.

2.     Provide significant words which are more relevant to the work in a logical sequence as ‘keywords’.

3.     Pay attention to the correct wording of the nomenclature. Desmodesmus sp. – "sp." no italics!

4.     The "Introduction" section should follow the state of the art of this field and review what has been done, for supporting the research gap and the significance of this study. Please improve the state of the art overview, to clearly show the progress beyond the state of the art.

5.     At the end of the introduction, the statement of the paper's goal and the explanation of novelty has to be properly formulated. Currently, this is not performed well. A high-quality paper has to provide a proper state-of-the-art analysis after the literature review and only based on the analysis to formulate the paper goals. The scientific basis and hypothesis for this study should be demonstrated in the "Introduction" section.

6.     The introduction of the review paper must be extended and reformulated in order to provide a more comprehensive approach. In the introduction, it is worth mentioning the optimization of the production and use of microalgae: https://doi.org/10.3390/pr8050517, https://doi.org/10.1016/j.biortech.2022.127014, https://doi.org/10.3390/cells11071206, https://doi.org/10.3390/en15082912, https://doi.org/10.3390/en15155713

7.     Statistical research is very important in experiments. For example, the hypothesis on distribution of each analyzed variable can be verified with a Shapiro-Wilk test. One-way analysis of variance (ANOVA) can be used to determine the significance of the difference between variables. Variance homogeneity in groups can be checked with Levene's test, whereas the significance of differences between the analyzed variables can be determined with a Tukey HSD test. If it has been tested, complete the methodology.

8.     Please indicate the manufacturer, city, country when mentioning the equipment e.g. line 249 Sigma Aldrich, line 289 Agilent GC-MS 5799A.

9.     How many repetitions was the test performed in? Laboratory tests should be checked. If they are, add standard deviations in the results.

10.  More details should be included in the conclusion.

11.  It is also recommended to discuss and explain what should be the appropriate policies based on the findings of this study.

Author Response

Reviewer 2

The authors thank Reviewer 2 for his comments and suggestions. The points will be answered thereafter, and eventual modifications added in the new submission.

  1. Abstracts needs to have more precision as in the current form it appears. In the abstract, please add an indication of the achievements from your study that is relevant to the journal scope. Please also add a research background.

The Abstract was revised, including some key features of our work:

“Thus, the extraction with hexane proved to be an effective way of removing nitrogen compounds, undesirable in fuels. The estimated low heating values of the present C5-C12 pyrolysis hydrocarbon fractions (between 43 and 44 MJ/kg) are quite comparable to the reported values for reformulated and conventional industrial gasolines (42 and 43 MJ/kg, respectively).”

  1. Provide significant words which are more relevant to the work in a logical sequence as ‘keywords’.

We changed the keywords, and their order. Now they are:

Keywords: Desmodesmus sp. microalgae; fast catalytic pyrolysis; n-hexane extract; biogasoline fraction”

  1. Pay attention to the correct wording of the nomenclature. Desmodesmus sp. – "sp." no italics!

In all the new submitted work, the writing “Desmodesmus sp.” was used.

  1. The "Introduction" section should follow the state of the art of this field and review what has been done, for supporting the research gap and the significance of this study. Please improve the state-of-the-art overview, to clearly show the progress beyond the state of the art.

In the Introduction section, we reinforced some key points, like:

“Microalgae transformation is therefore an impressive and challenging line of research implying progress in strain selection, in production technology, harvesting and post-harvesting, as well as microalgae transformation processes to increase renewable feed sources potential and gain independence against fossil ones.”

And we stressed maybe the most important one, where our contribution is focused, that is, the reduction of nitrogen compounds done by hexane extraction; some review articles have been already cited, refs. [10-14]:

“The extraction of microalgae oily molecules is effective in the simplification of the matrix and reduces the content of nitrogen compounds [10-14], although reduces the overall yield of fuel.”

  1. At the end of the introduction, the statement of the paper's goal and the explanation of novelty has to be properly formulated. Currently, this is not performed well. A high-quality paper has to provide a proper state-of-the-art analysis after the literature review and only based on the analysis to formulate the paper goals. The scientific basis and hypothesis for this study should be demonstrated in the "Introduction" section.

We now show more clearly our goals, the main one is the detailed comparison of pure biomass and biomass extract pyrolysis, in the C5-C12 range, and our contributions, the main one is the reduction of nitrogen compounds:

”The present study used both dried Desmodesmus sp. microalgae and its n-hexane extract, focusing on the production of bio-gasoline, i.e., the formation of C5-C12 hydrocarbons fraction. The catalytic pyrolysis was performed in presence of γ-Al2O3 as a reference catalyst, used in the past by different groups to efficiently deoxygenate oily molecules from other sources through catalytic pyrolysis [14, 22-25]. In the present work, extraction with hexane proved to be an effective way of removing undesirable nitrogen compounds in fuels.”

Another important result is now added in the text, and in the Abstract and Conclusion sections, that is the estimation of the heating values of these C5-C12 fractions (Tab. 4), whose values are quite similar to commercial gasolines.

  1. The introduction of the review paper must be extended and reformulated in order to provide a more comprehensive approach. In the introduction, it is worth mentioning the optimization of the production and use of microalgae: https://doi.org/10.3390/pr8050517 , https://doi.org/10.1016/j.biortech.2022.127014 , https://doi.org/10.3390/cells11071206 , https://doi.org/10.3390/en15082912 , https://doi.org/10.3390/en15155713

We now added a new sentence in the text, in the 85-91 lines, discussing about this aspect of the subject related to our research, adding three more references in the paper:

“Microalgae strains are important in wastewater treatment [15-17], and among them, Desmodesmus sp. emerged as the most promising strain, removing almost all nitrogen and phosphate from effluents. Its biomass presented a calorific value similar to terrestrial plants, and also had potential for use as a bio-lubricant. Therefore, Desmodesmus sp. shows promise for wastewater treatment, energy, and industrial applications. [18]; if such an application progresses, more Desmodesmus microalgae will be disponible as biomass feed for energy purpose.”

References:

[15] Barros, R.; Raposo. S.; Morais, E.G.; Rodrigues, B.; Afonso, V.; Gonçalves, P.; Marques, J.; Cerqueira, P.R.; Varela, J.; Teixeira, M.R.; Barreira, L. Biogas Production from Microalgal Biomass Produced in the Tertiary Treatment of Urban Wastewater: Assessment of Seasonal Variations. Energies 2022, 15(15), 5713. https://doi.org/10.3390/en15155713

[16] Zhao, G.; Wang, X.; Hong, Y.; Liu, X.; Wang, Q.; Zhai, Q.; Zhang, H. Attached cultivation of microalgae on rational carriers for swine wastewater treatment and biomass harvesting. Bioresour. Technol. 2022, 351, 27014. https://doi.org/10.1016/j.biortech.2022.127014

[17] Zieliński, M.; Dębowski, M.; Kazimierowicz, J. Outflow from a Biogas Plant as a Medium for Microalgae Biomass Cultivation—Pilot Scale Study and Technical Concept of a Large-Scale Installation. Energies 2022, 15(8), 2912; https://doi.org/10.3390/en15082912

  1. Statistical research is very important in experiments. For example, the hypothesis on distribution of each analyzed variable can be verified with a Shapiro-Wilk test. One-way analysis of variance (ANOVA) can be used to determine the significance of the difference between variables. Variance homogeneity in groups can be checked with Levene's test, whereas the significance of differences between the analyzed variables can be determined with a Tukey HSD test. If it has been tested, complete the methodology.

We didn’t perform ANOVA’s tests, but we’re always very careful about reliability of the data we’re producing in our laboratory; we now added in the text two new paragraphs and one Table including this subject, in the Methodology Section:

“The uncertainty of measurements in the pyrolysis experiments as a whole, including from sample preparation to chromatographic analysis, was evaluated by running duplicates, performed by the same operator, and the average percent standard deviation for some classes of compounds was calculated, as shown in Table 5 below:

Table 5. Standard deviation calculated for some types of hydrocarbons detected in the pyrolysis experiments.

Type of hydrocarbon

Standard Deviation (%)

Alkane

7.3

Alkene

2.7

Aromatic

2.1

General average

5.7

The results of the standard deviations obtained for these specific types of compounds demonstrate a high reliability in the results presented, especially considering that the work involves a biomass.”

  1. Please indicate the manufacturer, city, country when mentioning the equipment e.g. line 249 Sigma Aldrich, line 289 Agilent GC-MS 5799A.

The reviewer’s suggestion was attended.

  1. How many repetitions was the test performed in? Laboratory tests should be checked. If they are, add standard deviations in the results.

Please see the answer of question Nr. 7

  1. More details should be included in the conclusion.

We added new findings in the Conclusion Section, highlighted in bold characters below:

“4. Conclusions

The present study showed that the fast pyrolysis at 600°C of Desmodesmus sp. microalgae or/and its n-hexane extract, in absence or in presence of transition alumina catalyst, can be considered a way to produce a precursor of “drop in” gasoline. The alumina catalyst increased the quantity of C5-C12 hydrocarbon compounds when compared to purely thermal pyrolysis representing about 40% of all the dry microalgae pyrolysis products, and 33.5% of the products when using the n-hexane extract. A detailed analysis showed that linear molecules, mainly unsaturated, are predominant in both cases. Dry biomass formed more aromatics but less cyclic and alkylated molecules in relation to the n-hexane extract. Extraction with hexane proved to be an effective way of removing undesirable nitrogen compounds in fuels, as nitrogenated products, essentially alkylated pyrroles, were produced in a rather large extent with dry biomass, but were under the detection limit when pyrolyzing the extracts. Compared to industrial pyrolysis gasoline, the C5-C12 fraction obtained always contained a quite limited quantity of C8-C9 molecules. This “drop in” gasoline fraction is not fully comparable to industrial pyrolysis gasoline and its composition could probably be optimised using other types of catalysts to form more isomerized and hydrogenated products at the end of pyrolysis without increasing too much the formation of aromatics. Despite this fact, the estimated low heating values of the present C5-C12 pyrolysis hydrocarbon fractions (between 43 and 44 MJ/kg) are quite comparable to the reported values for reformulated and conventional industrial gasolines (42 and 43 MJ/kg, respectively). Selection of microalgae containing a higher content of oily components, with less poly-unsaturated compounds, is probably another way to optimize further the quality of biogasoline issued from fast pyrolysis of microalgae. Strategies to reduce the content of nitrogen compounds, such as the one presented here, is also another objective to increase the quality of microalgae pyrolysis products

  1. It is also recommended to discuss and explain what should be the appropriate policies based on the findings of this study.

In the end of the Results and Discussion Section, we added:

“Other topics that could be the subject of future studies include extraction using green solvents, preliminary thermal treatments such as torrefaction to limit the amount of nitrogenated compounds in dry biomass, and the use hydrogen assisted pyrolysis. Regarding the reaction itself, metal catalysts supported on alumina, and the possibility of composite or multi-layer catalyst bed could be exploited in the future.”

And in the end of the Conclusions Section, some policies have been already indicated, and now we added another one:

“Selection of microalgae containing a higher content of oily components, with less poly-unsaturated compounds, is probably another way to optimize further the quality of biogasoline issued from fast pyrolysis of microalgae. Strategies to reduce the content of nitrogen compounds, such as the one presented here, is also another objective to increase the quality of microalgae pyrolysis products.”

Round 2

Reviewer 1 Report

The authors revised the manuscript carefully based on the reviewers' comment. It can be published in Catalysts as it is.

Reviewer 2 Report

Thank you for considering my comments.